# Mycetoma as a major cause of limb amputation in Northeastern Ethiopia: A facility-based retrospective study

Wendemagegn Enbiale[1,2]*, Borna Nyaoke[3], Alemayehu Bekele[2], Kedir Ahmed Mohammed[4,5], Dereje Bedane[1]

**1** Department of Dermatovenereology College of Medicine and Health Sciences, Bahir Dar University, Bahir Dar, Ethiopia, **2** Collaborative Research and Training Center for Neglected Tropical Diseases, Arba Minch University, Arba Minch, Ethiopia, **3** Drug for Neglected Disease, Nairobi, Kenya, **4** Department of Medicine, School of Medicine, Samara University, Samara, Ethiopia, **5** Dubti General Hospital, Afar Region, Ethiopia

* Wendemagegnenbiale@gmail.com

## Abstract

### Background

Mycetoma is a chronic, progressively destructive infection that can result in severe disability and limb loss. In Ethiopia, diagnostic capacity and access to effective treatment remain limited, and the burden of mycetoma is poorly characterized. Recent clinical observations from the Afar Region suggest a high frequency of advanced disease and amputation, yet systematic evidence on the burden is lacking. This study aimed to describe the clinical burden of mycetoma, diagnostic and treatment practices, care-seeking patterns, and the extent of limb amputation at Dubti General Hospital in Afar region, northeastern Ethiopia. We conducted a facility-based retrospective review of all patients with a clinical diagnosis of mycetoma managed at Dubti General Hospital between September 2020 and August 2025. Demographic characteristics, clinical presentation, diagnostic investigations, treatment modalities, disability status, and surgical outcomes were summarized descriptively. Factors associated with delayed presentation (>12 months from symptom onset) were assessed using multivariable logistic regression. A total of 143 patients were identified, with a mean age of 30.9 years (SD ± 11.7); 79% were male, 85.3% resided in rural areas, and 46% were pastoralists. All cases involved the lower limb and presented with localized swelling. Pain (90.9%), warmth (54.5%), sinus formation (42.7%), and discharge (40.6%) were common. Diagnosis relied primarily on clinical assessment alone (58.7%), with limited use of imaging and biopsy. The mean duration of illness before first presentation was 33.8 months (SD ± 29), and 89.5% of patients presented after more than 12 months of symptoms. Compared with farmers, merchants had lower odds of delayed presentation (AOR = 0.89, 95% CI: 0.27–0.59). Nineteen patients (13.3%) underwent limb amputation, accounting for 23.5% of all orthopaedic

which permits unrestricted use, distribution, and reproduction in any medium, provided the original author and source are credited.

**Data availability statement:** All data are in the manuscript and/or supporting information files.

**Funding:** The author(s) received no specific funding for this work.

**Competing interests:** The authors have declared that no competing interests exist.

amputations performed at the hospital during the study period. Disability at presentation was frequent, with 14.0% of patients experiencing severe motor impairment. Mycetoma in Afar predominantly affects young rural men and presents almost exclusively with advanced lower-limb disease. Profound diagnostic limitations, delayed care-seeking, and restricted surgical options contribute to poor outcomes. Integrating mycetoma into national neglected tropical disease strategies, strengthening early detection and diagnostic services, ensuring consistent access to essential medications, and expanding limb-sparing surgical capacity are critical to reducing preventable disability and aligning Ethiopia's response with global NTD control targets.

## Author summary

Mycetoma is a long-lasting and destructive infection that mainly affects people living in hot, rural areas. It usually begins as a painless swelling of the foot after bacteria or fungi enter the skin through small injuries, such as thorn pricks. When diagnosis and treatment are delayed, the infection can spread to deeper tissues and bone, often leaving amputation as the only life-saving option. Although mycetoma is recognized as a neglected tropical disease, its contribution to limb loss in Ethiopia has not been well documented. In this study, we reviewed medical records of patients diagnosed with mycetoma at Dubti Referral Hospital in the Afar Region of northeastern Ethiopia. We identified 143 patients, most of whom were young men from rural pastoralist communities. Nearly all patients presented with advanced disease of the lower limb after living with symptoms for several years. Diagnostic services were limited, with most cases diagnosed clinically and few receiving imaging or laboratory confirmation. As a result, many patients developed severe disability, and mycetoma accounted for almost one quarter of all limb amputations performed at the hospital during the study period. Our findings show that mycetoma is a major and preventable cause of limb amputation in Afar. Late presentation, limited diagnostic capacity, and lack of access to limb-sparing treatment contribute to unnecessary disability and loss of livelihood. Improving early detection, expanding access to imaging and laboratory testing, and strengthening surgical and medical management could substantially reduce amputations and protect the lives and productivity of people affected by mycetoma in Ethiopia and other endemic settings.

## Introduction

Mycetoma is a chronic, progressively destructive infection of the skin and subcutaneous tissues caused by filamentous bacteria (actinomycetoma) or fungi (eumycetoma) [1]. It classically manifests as a triad of a painless subcutaneous mass, multiple sinus tracts, and the discharge of grains containing the causative organism [2]. The disease primarily affects the foot or hand and evolves slowly over years, often leading to massive local destruction, deformity, and sometimes limb loss if untreated [3].

Mycetoma thrives in impoverished rural settings of tropical and subtropical regions, collectively known as the "mycetoma belt," which spans countries such as Sudan, Ethiopia, India, Mexico, and Yemen [1,4]. Its epidemiology is shaped by climate (typically arid or semi-arid), socioeconomic vulnerability, and occupational exposures such as farming and herding without protective footwear [5]. Males aged 20–40 are disproportionately affected due to their occupational risk profiles [6]. In recognition of its disabling consequences and neglected status, mycetoma was designated as a Neglected Tropical Disease (NTD) by the World Health Organization (WHO) in 2016 [1].

Despite its inclusion on the WHO NTD list, reliable global data on mycetoma burden remain limited due to underreporting, lack of mandatory notification systems, and poor diagnostic infrastructure [1,4]. A systematic review estimated ~8,763 cases globally, with the highest case volumes reported from Sudan, India, and Mexico [4]. Sudan has long been considered a mycetoma hotspot, where prevalence in some endemic villages reaches up to 8.5 per 1,000 inhabitants [5]. In contrast, even in similarly at-risk countries like Ethiopia, the disease has only recently begun to receive focused research attention [6].

Until recently, Ethiopia lacked formal surveillance, laboratory capacity, and policy focus on mycetoma [6,7]. The first published case series, seven patients from Boru Meda Hospital, highlighted diagnostic delays (median disease duration of 5 years), clinical uncertainty (fungal vs. bacterial etiology undifferentiated), and empirical combination therapy [7]. A subsequent multi-site retrospective review (2018–2022) involving 13 referral hospitals identified 143 cases of subcutaneous mycoses, 82.5% of which were classified as mycetoma [6]. Cases were geographically widespread but disproportionately concentrated in northern Ethiopia, particularly the Tigray and Amhara regions. The typical patient was a young male agricultural worker, demographics matching global patterns [6,7]. However, the reported national case count remains low for a country of over 120 million, suggesting significant under detection.

Our recent unpublished clinical experience from Ethiopia's Afar region, particularly Dubti General Hospital, suggest a higher clinical burden than previously recognized. Hospital reports identified mycetoma as a leading indication for limb amputation in the region. However, systematic studies examining outcomes or predictors of amputation are lacking. Similar gaps exist across sub-Saharan Africa, where most published literature comprises case reports or small descriptive series. By contrast, large-scale research from endemic countries like Sudan and Mexico has provided valuable insights into treatment outcomes and risk factors for limb loss [8,9]. For example, a study at Sudan's national referral center reported that only 6% of mycetoma patients underwent amputation, with fungal infections and recurrent disease being major risk factors [9].

In Ethiopia, the current literature is hampered by several limitations. First, most studies have small sample sizes drawn from tertiary facilities, introducing referral bias [6,7]. Second, diagnostic capacity is inconsistent; many cases lack microbiological or imaging confirmation, making etiological classification difficult [7,8]. Third, treatment practices vary widely and often include empirical regimens due to a lack of diagnostics. Most critically, no Ethiopian study has comprehensively examined the surgical intervention, including the prevalence, types, and predictors of limb amputation.

To address this urgent evidence gap, the present study uses real-world clinical data from northeastern Ethiopia to characterize mycetoma-related amputations. Specifically, we aim to (i) quantify the prevalence of limb amputation among patients diagnosed with mycetoma, (ii) describe the types and anatomical levels of amputation performed, (iii) assess clinical and demographic factors associated with increased risk of limb loss, and (iv) document early post-operative outcomes and complications. To our knowledge, this is the first systematic investigation of mycetoma-related amputations in Ethiopia, and it will provide essential data for national health planning, mycetoma control strategies, and clinical practice optimization in other endemic settings.

## Methods

### Ethics approval

Ethical approval for secondary analysis of clinical registry data was obtained from Arba Minch University College of Medicine and Health Science, Ethical Review Board on July 15th, 2025, Ref. No. IRB/23359/25. A support letter was secured from Dubti General Hospital. As the study used de-identified routine data, informed consent was waived in accordance with national research guidelines.

## Study design

We conducted a facility-based retrospective observational study using routinely collected clinical data from a patient files of individual diagnosed as mycetoma from September 2020 to August 2025.

## Setting

The study was conducted at Dubti General Hospital, the main referral care centre in the Afar National Regional State of northeastern Ethiopia. Afar is a predominantly pastoralist region bordered by Eritrea to the northeast and Djibouti to the east, with an estimated population of 1.8 million, of whom 54.7% are male and 45.3% female. The regional capital, Samara, lies approximately 600 km from Addis Ababa, and Dubti town, where the hospital is located, is 10–12 km from Samara, the capital of the regional state. Dubti General Hospital serves as the main referral facility for Zone 1 and much of the wider Afar Region, providing care for an estimated catchment population of nearly one million people. The hospital has multiple inpatient and outpatient service units; the inpatient service unit includes medical, surgical, paediatrics, gynae-cology, and Oncology ward, while the outpatient service unt includes emergency, maternal and child health, antiretroviral therapy clinic and cold OPD.

With more than 400 health-care workers, including physicians, nurses, midwives, laboratory professionals, and phar-macy staff, Dubti Hospital provides comprehensive services in a resource-limited, arid environment where access to specialized diagnostics is constrained and patients often present with advanced disease.

## Sample population

All patients with a clinical diagnosis of mycetoma who have been seen and treated during the study period were eligible for inclusion.

Functional impairment at presentation was categorized based on documentation in clinical records as:

- Mild impairment: localized swelling without documented bone involvement and no limitation in ambulation;

- Moderate impairment: documented bone involvement and/or mobility limitation without major deformity;

- Severe impairment: marked deformity, extensive bone destruction, or major limitation in mobility.

### Inclusion criteria:.

1. Clinical diagnosis of mycetoma (eumycetoma or actinomycetoma) based on the presence of swelling, sinuses, and/or grains.

2. Availability of complete clinical records, including demographic information, lesion characteristics, and treatment history.

### Exclusion criteria:.

1. Records with missing key outcome variables.

2. Uncertain diagnosis or alternative confirmed cause of subcutaneous swelling.

## Data collection

Data were extracted from the hospital's outpatient registry and patient charts, including standardized clinical assessment forms used during routine follow-up. The dataset was exported from the registry into a structured electronic format and cleaned for internal consistency, duplication, and missingness prior to analysis. Where available, laboratory, imaging, and surgical notes were cross-checked to validate clinical findings.

## Variables

### Primary outcome.

- Amputation status, defined as any recorded surgical removal of part or all of a limb attributable to mycetoma. Amputation was inferred from structured fields specifying amputation type (toe, partial foot amputation, ankle disarticulation, below-knee amputation, above-knee amputation, knee disarticulation, hip disarticulation).

### Independent variables:.

- Demographic: age, sex, residence.
- Clinical: lesion site, presence of sinuses, grains, swelling, bone involvement (X-ray), duration of illness (proxied by age at diagnosis).
- Management-related: prior surgical treatment, availability and type of diagnostic investigations (microscopy, imaging).
- Candidate predictors of amputation included bone involvement (X-ray documented), duration of illness, prior treatment failure, documented recurrence, and severe functional impairment.

## Methodological limitation

Although the initial objective was to retrieve five years of retrospective data, incomplete retrieval of earlier paper-based files limited the study period to September 2022–August 2025, during which registry documentation was consistently available. In the original project proposal, we also aimed to document early post-operative outcomes and complications; however, this information was not available in the patient records.

## Analysis strategy

Data were analysed using descriptive and inferential statistics. Categorical variables were summarized using frequencies and percentages, and continuous variables using means, medians, or interquartile ranges. Bivariate comparisons between amputated and non-amputated patients used $\chi^2$ tests or Wilcoxon rank-sum tests where appropriate. Logistic regression was performed to assess predictors of amputation, adjusting for demographic and clinical covariates. Findings were reported using adjusted odds ratios (AORs) with 95% confidence intervals.

No formal sample size calculation was applied because all eligible patients within the registry were included.

## Results

### Socio-demographic characteristics

A total of 143 patients with clinically diagnosed mycetoma were found registered in three years period. The mean age of patients was 30.9 years (SD ± 11.7). Most were male (79%) and residents of rural areas (85.3%). Pastoralists accounted for 46% of cases, followed by farmers (28.7%) and housewives (12%) (Table 1).

### Clinical characteristics of mycetoma patients

All patients (100%) presented with localized swelling, and all lesions involved the lower limb. Seventy one patient had involvement of the right foot, 70 of left foot and 2 both feet involved. Other commonly documented signs and symptoms included:- pain (90.9%), warmth of lesion (54.5%), sinus formation (42.7%), discharge (40.6%) and itching (19.6%). (Table 2). Grain was documented only from 3.5% of record, limb deformity recorded in 2.8%. All diagnosed cases involved the lower limb.

**Table 1. Socio-Demographic Characteristics of Mycetoma Patients at Dupti General Hospital.**

|  | Category | Frequency | Percentage |
|---|---|---|---|
| **Sex** | Female | 30 | 21 |
|  | Male | 113 | 79 |
| **Residence** | Rural | 122 | 85.3 |
|  | Urban | 21 | 14.7 |
| **Occupation** |  |  |  |
|  | Farmer | 43 | 28.7 |
|  | Pastoralist | 69 | 46 |
|  | Housewife | 18 | 12 |
|  | Merchant | 5 | 3.3 |
|  | Other | 1 | 0.7 |
|  | Student | 14 | 9.3 |

**Table 2. Clinical features of Mycetoma patients at Dupti referral hospital.**

| Variable | Category | Frequency | Percent |
|---|---|---|---|
| Localized swelling | Yes | 143 | 100 |
|  | No | 0 | 0 |
| Opening on the skin(sinuses) | Yes | 61 | 42.7 |
|  | No | 14 | 9.8 |
|  | Not mentioned | 68 | 47.6 |
| Discharge | Yes | 58 | 40.6 |
|  | No | 19 | 13.3 |
|  | Not mentioned | 66 | 46.2 |
| Grain | Yes | 5 | 3.5 |
|  | No | 4 | 2.8 |
|  | Not mentioned | 134 | 93.7 |
| Limb deformity | Yes | 4 | 2.8 |
|  | No | 1 | 0.7 |
|  | Not mentioned | 128 | 89.5 |
| Itching | Yes | 28 | 19.6 |
|  | No | 1 | 0.7 |
|  | Not mentioned | 101 | 70.6 |
| Hotness/ warmth of lesion | Yes | 78 | 54.5 |
|  | No | 2 | 1.4 |
|  | Not mentioned | 63 | 44.1 |
| Pain | Yes | 130 | 90.9 |
|  | No | 13 | 9.1 |

## Diagnostic practice

More than half of patients (58.7%) were diagnosed using clinical assessment alone. Imaging modalities used included:

- X-ray: 38.5%

- Ultrasound: 2.8%

- CT scan: 0.7%

Biopsy was performed in 21 patients (14.7%), with etiological identification without species level identification was achieved in 9 cases: 3 actinomycetoma and 6 eumycetoma (Table 3).

## Health-care seeking and associated factors

The mean duration from onset of swelling to first presentation was 33.8 months (SD ± 29). A large majority (89.5%) presented after more than 12 months symptom onset. In multivariable logistic regression, occupation was the only variable significantly associated with delayed presentation. Compared with farmers (reference group), merchants had significantly lower odds of delayed presentation (AOR = 0.89, 95% CI: 0.27–0.59, p = 0.0181). No statistically significant associations were observed for sex, residence, age, or other occupations (Table 4).

## Treatment practice

Among all patients, 123 received medical therapy, 12 underwent surgery alone, and 8 received both medical and surgical treatment. Medical management primarily involved combination antifungal therapy (fluconazole + cotrimoxazole or ketoconazole + cotrimoxazole).

Surgical management was predominantly limb amputation. A total of 19 patients underwent amputation, representing 23.5% of all orthopedic amputations at Dubti Referral Hospital during the study period (19/81) (Fig 1).

**Table 3. Diagnostic Practices Among Mycetoma Patients at Dubti Referral Hospital (n = 143).**

| Diagnostic Method | Number (n) | Percentage (%) |
|---|---|---|
| No imaging performed (clinical diagnosis only) | 84 | 58.7 |
| X-ray imaging | 55 | 38.5 |
| Ultrasound | 4 | 2.8 |
| CT scan | 1 | 0.7 |
| Biopsy performed | 21 | 14.7 |
| Etiology identified (bacterial Vs fungal) | 9 | 42.9* |
| - Actinomycetoma | 3 | — |
| - Eumycetoma | 6 | — |

*Percentage calculated among those who received a biopsy (9/21).

**Table 4. Binary Logistic regression analysis showing the factors associated with delay presentation of patients to health facility after onset of swelling at Dupti referral hospital.**

| Variable | Category | Presentation | | AOR | 95% CI | P-value |
|---|---|---|---|---|---|---|
| | | Delayed | Not delayed | | | |
| **Age** | Age | | | 3.59 | 1.03, 29.9 | 0.34 |
| **Sex** | Female | 29 | 1 | Ref | Ref | Res |
| | Male | 99 | 14 | 1.33 | 0.2, 2.89 | 0.38 |
| **Residence** | Rural | 111 | 11 | Ref | Ref | Res |
| | Urban | 17 | 4 | 2.9 | 0.65, 4.6 | 0.77 |
| **Occupation** | Farmer | 40 | 4 | Ref | Ref | Res |
| | housewife | 18 | 0 | – | – | – |
| | Merchant | 2 | 3 | 0.89 | 0.27, 0.59 | 0.0181* |
| | Pastoralist | 57 | 5 | 3.53 | 0.78, 3.96 | 0.95 |
| | Student | 11 | 3 | 0.54 | 0.17, 1.48 | 0.103 |

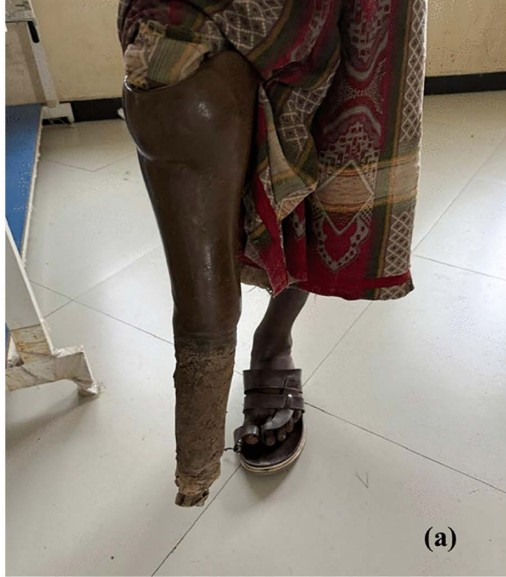 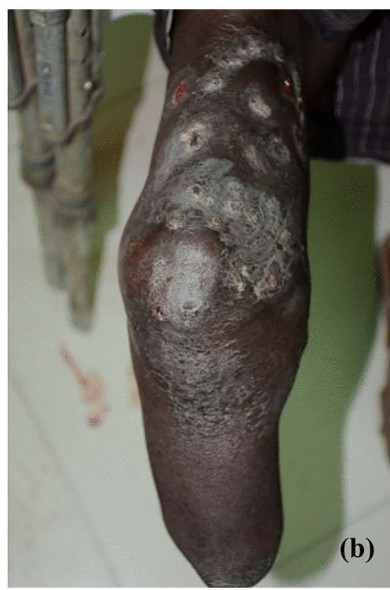

**Fig 1. (a) A 40-year-old male patient with a below-knee amputation of the right leg following a diagnosis of mycetoma.** (b) A 34-year-old male patient with a below-knee amputation complicated by recurrence of mycetoma at the amputation stump. Both patients were diagnosed and underwent amputation at Dubti Hospital, Afar Ethiopia.

## Disability

At presentation, severe motor impairment was documented in 20 (14.0%) patients, moderate impairment in 29 (20.3%) patients, and mild impairment in 94 patients (65.7%). Severe impairment was typically associated with marked bone destruction or deformity, while mild impairment showed no bone involvement.

## Discussion

This study provides one of the most detailed assessments to date of the clinical burden, diagnostic practices, treatment patterns, and disability associated with mycetoma in Ethiopia's Afar Region. Among 143 clinically diagnosed patients, we found that the disease primarily affects young, rural men engaged in pastoralist or agricultural livelihoods, consistent with global and regional evidence that occupational exposure and environmental conditions shape mycetoma risk [1–4]. Almost all patients presented with long-standing disease involving the lower limb, and most exhibited symptoms indicative of advanced progression, including pain, sinus formation, discharge, or warmth. These findings demonstrate the significant burden of mycetoma in Afar and highlight how late presentation contributes to disability and surgical intervention.

Diagnostic capacity was notably limited. More than half of patients were diagnosed clinically without any imaging, and only a small proportion received X-ray, ultrasound, or CT evaluation. Histopathology or etiological classification was rarely achieved, with only 9 confirmed cases across all biopsies. These gaps align with earlier Ethiopian reports describing restricted access to imaging and laboratory support, resulting in diagnostic uncertainty and delayed management [6–8]. In contrast, centres in Sudan and Mexico, where mycetoma services are more established, routinely use imaging and laboratory confirmation to guide treatment decisions and reduce complications [8–9]. Strengthening diagnostic capacity, therefore, represents a critical step in improving outcomes in Afar.

Delayed health-care seeking was pervasive, with nearly 90% of patients presenting after more than 12 months of symptoms. Only the merchant occupation was associated with earlier presentation, likely reflecting greater mobility or

access to resources. Similar patterns of delayed care have been documented in other endemic regions and are typically driven by low community awareness, long distances to health facilities, and limited availability of diagnostic services [1,3,6]. Given that late presentation correlates strongly with bone involvement and disability, community-based education and early referral pathways should be prioritized.

The burden of disability and amputation in this cohort was substantial. Severe motor impairment affected a notable proportion of patients, and 19 individuals required amputation, representing nearly one quarter of all orthopaedic amputations performed during the study period. This amputation rate is considerably higher than that reported from Sudan's Mycetoma Research Centre, where approximately 6% of cases require amputation due to earlier diagnosis and access to limb-sparing interventions [8]. Ethiopia's current lack of standardized national guidelines and limited surgical expertise in mycetoma likely contribute to the higher proportion of advanced cases requiring major surgery [5–7]. Notably, no limb-sparing surgical procedures such as debridement were documented in this cohort, underscoring critical gaps in surgical capacity.

Several strengths enhance the value of this study, including the use of a relatively large clinical cohort from a high-burden region and the inclusion of detailed diagnostic and treatment data from a referral hospital serving remote pastoralist communities. This study has several limitations. As a retrospective review of routine clinical documentation, it relied on the completeness and accuracy of medical records, which were variable. Detailed information on diagnostic workflows, duration of medical therapy, and indications for specific management decisions was not consistently available. Key clinical indicators such as imaging findings, extent of bone involvement, and grain characteristics were frequently missing. Histopathological confirmation of amputated specimens and etiological classification were rarely documented, limiting robust comparisons between actinomycetoma and eumycetoma. In addition, the intended five-year retrospective period could not be fully achieved due to incomplete retrieval of older paper-based records. Finally, because the study included only patients who reached a tertiary referral hospital, the findings may underestimate the true community-level burden of mycetoma and related amputations in the region. Despite these limitations, the findings have important implications for clinical practice, regional health planning, and national policy development in Ethiopia. First, the high proportion of advanced disease and amputations underscores the urgent need to integrate mycetoma into Ethiopia's NTD surveillance and reporting systems. Formal inclusion within routine health information systems would improve case detection, resource allocation, and monitoring of outcomes.

Second, strengthening early diagnosis at primary and district health facility levels is critical. This could be achieved through targeted training of frontline health workers in endemic regions, development and dissemination of standardized national diagnostic and treatment guidelines, and expansion of access to basic imaging (particularly X-ray) at regional hospitals. Establishing clear referral pathways from health centres to referral facilities would further reduce delays in care.

Third, the substantial burden of amputation highlights the need to build regional surgical capacity for limb-sparing interventions and to ensure consistent availability of essential antifungal and antibacterial medications. Developing standardized treatment protocols and ensuring drug supply chains through national procurement systems would reduce reliance on empirical or inconsistent management approaches.

Finally, prevention and early detection strategies should be embedded within existing regional NTD and community health programs. In pastoralist and agricultural communities, health education campaigns promoting early care-seeking, protective footwear use, and awareness of early symptoms could substantially reduce progression to bone involvement and disability.

In conclusion, mycetoma remains a significant cause of morbidity, disability, and amputation in the Afar Region of Ethiopia.

The predominance of advanced disease at presentation reflects delayed care-seeking, limited diagnostic capacity, and the absence of standardized management pathways. This study provides important real-world evidence demonstrating the clinical and surgical burden of mycetoma in an endemic setting and highlights critical gaps in early detection and care.

Addressing these gaps requires coordinated action at multiple levels, including integration of mycetoma into national NTD strategies, strengthening of diagnostic and surgical capacity at regional hospitals, development of standardized clinical guidelines, and expansion of community-based early detection initiatives. Without such interventions, preventable disability and limb loss will continue to disproportionately affect rural and pastoralist populations.

Investments in early detection, diagnostic strengthening, and accessible, standardized management pathways are urgently needed to prevent progression to irreversible disability and to align Ethiopia's response with global targets for neglected tropical diseases.

## Supporting information

**S1 Data. A cleaned dataset for Mycetoma as Cause of Amputation Clean data cleaned up.**
(XLSX)

## Author contributions

**Conceptualization:** Wendemagegn Enbiale, Dereje Bedane.

**Data curation:** Wendemagegn Enbiale, Alemayehu Bekele, Dereje Bedane.

**Formal analysis:** Wendemagegn Enbiale, Dereje Bedane.

**Funding acquisition:** Wendemagegn Enbiale.

**Investigation:** Wendemagegn Enbiale, Alemayehu Bekele, Kedir Ahmed Mohammed, Dereje Bedane.

**Methodology:** Wendemagegn Enbiale, Kedir Ahmed Mohammed.

**Project administration:** Wendemagegn Enbiale, Kedir Ahmed Mohammed, Dereje Bedane.

**Supervision:** Wendemagegn Enbiale, Borna Nyaoke, Dereje Bedane.

**Validation:** Borna Nyaoke, Alemayehu Bekele, Kedir Ahmed Mohammed, Dereje Bedane.

**Visualization:** Wendemagegn Enbiale, Borna Nyaoke, Kedir Ahmed Mohammed, Dereje Bedane.

**Writing – original draft:** Wendemagegn Enbiale.

**Writing – review & editing:** Wendemagegn Enbiale, Borna Nyaoke, Alemayehu Bekele, Kedir Ahmed Mohammed, Dereje Bedane.

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
