## [Decision Letter · Decision Letter 0]

28 Jan 2026

PNTD-D-25-02277

Mycetoma as a Major Cause of Limb Amputation in Northeastern Ethiopia: A Facility-Based Retrospective Study

Dear Dr. Enbiale,

Thank you for submitting your manuscript to PLOS Neglected Tropical Diseases. After careful consideration, we feel that it has merit but does not fully meet PLOS Neglected Tropical Diseases's publication criteria as it currently stands. Therefore, we invite you to submit a revised version of the manuscript that addresses the points raised during the review process.

Please submit your revised manuscript within by Mar 29 2026 11:59PM. If you will need more time than this to complete your revisions, please reply to this message or contact the journal office at plosntds@plos.org. Please include the following items when submitting your revised manuscript:

We look forward to receiving your revised manuscript.

Kind regards,

Joshua Nosanchuk, MD

Section Editor

Shaden Kamhawi

co-Editor-in-Chief

Paul Brindley

co-Editor-in-Chief

Journal Requirements:

At this stage, the following Authors/Authors require contributions: Wendemagegn Enbiale, Borna Nyaoke, Alemayehu Bekele, Kedir Ahmed Mohammed, and Dereje Bedane. Please ensure that the full contributions of each author are acknowledged in the "Add/Edit/Remove Authors" section of our submission form.

2) Please insert an Ethics Statement at the beginning of your Methods section, under a subheading 'Ethics Statement'. It must include:

i) The full name(s) of the Institutional Review Board(s) or Ethics Committee(s)

ii) The approval number(s), or a statement that approval was granted by the named board(s)

iii) A statement that formal consent was obtained (must state whether verbal/written) OR the reason consent was not obtained (e.g. anonymity). NOTE: If child participants, the statement must declare that formal consent was obtained from the parent/guardian.].

Potential Copyright Issues:

i) Please confirm (a) that you are the photographer of 1, or (b) provide written permission from the photographer to publish the photo(s) under our CC BY 4.0 license.

5) Please send a completed 'Competing Interests' statement, including any COIs declared by your co-authors. If you have no competing interests to declare, please state "The authors have declared that no competing interests exist". Otherwise please declare all competing interests beginning with the statement "I have read the journal's policy and the authors of this manuscript have the following competing interests"

Reviewers' Comments:

Reviewer's Responses to Questions

Key Review Criteria Required for Acceptance?

Methods

-Are the objectives of the study clearly articulated with a clear testable hypothesis stated?

-Is the study design appropriate to address the stated objectives?

-Is the population clearly described and appropriate for the hypothesis being tested?

-Is the sample size sufficient to ensure adequate power to address the hypothesis being tested?

-Were correct statistical analysis used to support conclusions?

-Are there concerns about ethical or regulatory requirements being met?

Reviewer #1: To get more reliable data, suggestion to expand the study duration for 5 years

How other differential diagnosis of mycetoma e.g., malignancy and infections were ruled out?

What were the process for sending diagnostic tests? Does histopathology and culture of the amputed specimens were done to confirm the diagnosis?

Mention the duration of medical options as a variable in method?

Explain the severity impairment criteria in methods, how it was classified as sever, Moderate and mild.

Specify in the methods what were the clinical factors for limb loss? e.g., Bone involvement, massive and advance lesion, failed medical treatment, severe superimposed bacterial infections, patient disability, recurrence after surgery.

Results

-Does the analysis presented match the analysis plan?

-Are the results clearly and completely presented?

-Are the figures (Tables, Images) of sufficient quality for clarity?

Reviewer #1: No need for figure 2.

What were the pathogens that were positive in microbiology culture?

Elaborate on histopathological findings, e.g., involvement of bones and tissue reactions

Multivariate logistic regression - "merchant occupation was the only factor significantly associated with delayed care -seeking" This statement is contradictory to the results.

One of the objectives is to document early post-operative outcomes and complication, however in result no descriptions were available.

Conclusions

-Are the conclusions supported by the data presented?

-Are the limitations of analysis clearly described?

-Do the authors discuss how these data can be helpful to advance our understanding of the topic under study?

-Is public health relevance addressed?

Reviewer #1: Conclusion is appropriate, however, i suggest making the discussion more policy relevant that could help the Ethiopian ministry in policy development

Editorial and Data Presentation Modifications?

Reviewer #1: this paper will help in getting data from other regions of Ethiopia

Summary and General Comments

Reviewer #1: overall, this paper could help in developing any policy by getting the baseline data from different regions of Ethiopia. Improvement in clinical detail in the method sections would improve the quality of the paper.

PLOS authors have the option to publish the peer review history of their article (what does this mean?). If published, this will include your full peer review and any attached files.

Do you want your identity to be public for this peer review? For information about this choice, including consent withdrawal, please see our Privacy Policy.

Reviewer #1: Yes: Mohammad Zeeshan

Figure resubmission:
---

## [Editor Report · Decision Letter 1]

4 Mar 2026

PNTD-D-25-02277R1Mycetoma as a Major Cause of Limb Amputation in Northeastern Ethiopia: A Facility-Based Retrospective StudyPLOS Neglected Tropical Diseases Dear Dr. Enbiale, Thank you for submitting your manuscript to PLOS Neglected Tropical Diseases. We appreciate the thoughtful responses to the initial review; however, after careful consideration, we feel that it has merit but does not fully meet PLOS Neglected Tropical Diseases's publication criteria as it currently stands. Therefore, we invite you to submit a revised version of the manuscript that addresses the points raised during the review process. Please submit your revised manuscript by Apr 03 2026 11:59PM. If you will need more time than this to complete your revisions, please reply to this message or contact the journal office at plosntds@plos.org.  Please include the following items when submitting your revised manuscript:* A letter that responds to each point raised by the editor and reviewer(s). You should upload this letter as a separate file labeled 'Response to Reviewers'. This file does not need to include responses to any formatting updates and technical items listed in the 'Journal Requirements' section below.* A marked-up copy of your manuscript that highlights changes made to the original version. You should upload this as a separate file labeled 'Revised Manuscript with Track Changes'.* An unmarked version of your revised paper without tracked changes. You should upload this as a separate file labeled 'Manuscript'. If you would like to make changes to your financial disclosure, competing interests statement, or data availability statement, please make these updates within the submission form at the time of resubmission. Guidelines for resubmitting your figure files are available below the reviewer comments at the end of this letter. We look forward to receiving your revised manuscript. Kind regards, Joshua NosanchukSection EditorPLOS Neglected Tropical Diseases

Shaden Kamhawi

co-Editor-in-Chief

Paul Brindley

co-Editor-in-Chief

 Additional Editor Comments: Given that the response letter clearly states the study was "restricted to September 2022–August 2025" it is not appropriate to write in the abstract and methods that the study was over the 5 year 2020-2025 period. The authors should revise the manuscript to state the dates studied. It is reasonable to share in the methods that there was an approach for a longer period, but this was not achievable due to the lack of documentation at the center.

NTD should be written out in the abstract. It is appropriately defined in the main component of the document and can be used after that point.

Table 4. Needs clarification. The authors need to clarify what "Ref" and "Res" means as they are not defined under the table. Also, the results of the multivariate regression indicating that the farmers were the only significant factor are not shown in the text- the statistical result is not shown.

Please review for typos like

Table 1 having "House wife" and not the correct "Housewife" as well as Table 4 having "housewife" while the other occupations are all capitalized.

"out patient" should be "outpatient"

"In the project proposal also we aspired to" should be "we also aspired"

"clinical indicatorssuch as imaging findings, extent of bone involvement, and grain characteristics were frequently missing." Needs a space between indicators and such.

"Despite these limitations, the findings have important implications for clinical care. egional health planning, and national policy development in Ethiopia." The period after care should be a comma and the next word regional.  Journal Requirements:

1) We note that your Fig 1.tif, and Figure1 recurence of Mycetoma following amputation.tif files are duplicated on your submission. Please remove any unnecessary or old files from your revision, and make sure that only those relevant to the current version of the manuscript are included.

2) We have noticed that you have uploaded Supporting Information files, but you have not included a list of legends. Please add a full list of legends for your Supporting Information files after the references list.

**Reviewers' comments:**  **Figure resubmission:** While revising your submission, we strongly recommend that you use PLOS’s NAAS tool (https://ngplosjournals.pagemajik.ai/artanalysis) to test your figure files. NAAS can convert your figure files to the TIFF file type and meet basic requirements (such as print size, resolution), or provide you with a report on issues that do not meet our requirements and that NAAS cannot fix.

After uploading your figures to PLOS’s NAAS tool - https://ngplosjournals.pagemajik.ai/artanalysis, NAAS will process the files provided and display the results in the "Uploaded Files" section of the page as the processing is complete. If the uploaded figures meet our requirements (or NAAS is able to fix the files to meet our requirements), the figure will be marked as "fixed" above. If NAAS is unable to fix the files, a red "failed" label will appear above. When NAAS has confirmed that the figure files meet our requirements, please download the file via the download option, and include these NAAS processed figure files when submitting your revised manuscript. Reproducibility: To enhance the reproducibility of your results, we recommend that authors of applicable studies deposit laboratory protocols in protocols.io, where a protocol can be assigned its own identifier (DOI) such that it can be cited independently in the future. Additionally, PLOS ONE offers an option to publish peer-reviewed clinical study protocols. Read more information on sharing protocols at https://plos.org/protocols?utm_medium=editorial-email&utm_source=authorletters&utm_campaign=protocols

---

## [Editor Report · Decision Letter 2]

25 Mar 2026

PNTD-D-25-02277R2

Mycetoma as a Major Cause of Limb Amputation in Northeastern Ethiopia: A Facility-Based Retrospective Study

Dear Dr. Enbiale,

Although we appreciate your continued interest in publishing in our journal, the resubmission documents are sub-standard and incomplete. For example, there is no point-by-point response to the second submission, figure 4 is missing from the "manuscript" document, typos remain (example: missing period in the conclusions of abstract " poor outcomes Integrating mycetoma "), etc. However, given the value of your work, we will allow for a re-submission, but it is critical that this next version be very carefully reviewed prior to its return to our editorial staff.

Please submit your revised manuscript within by Apr 24 2026 11:59PM. If you will need more time than this to complete your revisions, please reply to this message or contact the journal office at plosntds@plos.org. Please include the following items when submitting your revised manuscript:

We look forward to receiving your revised manuscript.

Kind regards,

Joshua Nosanchuk, MD

Section Editor

Joshua Nosanchuk

Section Editor

Shaden Kamhawi

co-Editor-in-Chief

Paul Brindley

co-Editor-in-Chief

Reviewers' Comments:

Figure resubmission:
---

## [Editor Report · Decision Letter 3]

18 Apr 2026

Dear Dr Enbiale,

We are pleased to inform you that your manuscript 'Mycetoma as a Major Cause of Limb Amputation in Northeastern Ethiopia: A Facility-Based Retrospective Study' has been provisionally accepted for publication in PLOS Neglected Tropical Diseases.

Best regards,

Joshua Nosanchuk

Section Editor

Shaden Kamhawi

co-Editor-in-Chief

Paul Brindley

co-Editor-in-Chief

---

## [Editor Report · Acceptance letter]

Dear Dr Enbiale,

We are delighted to inform you that your manuscript, "Mycetoma as a Major Cause of Limb Amputation in Northeastern Ethiopia: A Facility-Based Retrospective Study," has been formally accepted for publication in PLOS Neglected Tropical Diseases.

Best regards,

Shaden Kamhawi

co-Editor-in-Chief

Paul Brindley

co-Editor-in-Chief
